# Does Ionized Magnesium Offer a Different Perspective Exploring the Association between Magnesemia and Targeted Cardiovascular Risk Factors?

**DOI:** 10.3390/jcm11144015

**Published:** 2022-07-11

**Authors:** Vanessa Gagliano, Fabian Schäffeler, Rosaria Del Giorno, Mario Bianchetti, Cesar Fabian Carvajal Canarte, José Joel Caballero Regueira, Luca Gabutti

**Affiliations:** 1Department of Internal Medicine, Clinical Research Unit, Regional Hospital of Bellinzona and Valli, Ente Ospedaliero Cantonale, 6500 Bellinzona, Switzerland; vanessa.gagliano@eoc.ch (V.G.); cesarfabian.carvajalcanarte@eoc.ch (C.F.C.C.); josejoel.caballeroregueira@eoc.ch (J.J.C.R.); 2Faculty of Biomedicine, Università della Svizzera Italiana, 6900 Lugano, Switzerland; fabian.schaeffeler@usi.ch (F.S.); rosaria.del.giorno@usi.ch (R.D.G.); mario.bianchetti@usi.ch (M.B.); 3Angiology Service, University Hospital of Lausanne, 1011 Lausanne, Switzerland; 4Department of Pediatrics, Regional Hospital of Bellinzona and Valli, Ente Ospedaliero Cantonale, 6500 Bellinzona, Switzerland

**Keywords:** ionized magnesium, total magnesium, pulse wave velocity (PWV), arterial stiffness, blood pressure, body mass index (BMI), body composition

## Abstract

Evidence of the association of magnesium (Mg) with arterial stiffness has so far been conflicting. The interplay between hypertension and elevated body mass index (BMI), with hypomagnesemia, instead, has been described in the literature in a more consistent way. Our study aims at revisiting the correlations between blood Mg levels and hemodynamic and body composition parameters in the general population, exploring the sensitivity profile of ionized Mg (Ion-Mg) compared to total Mg (Tot-Mg). We collected data from 755 subjects randomly chosen from a Swiss population previously described and stratified our sample into four equivalent classes according to ionized (whole blood) and total (serum) magnesium. After correcting for age, statistically significant differences emerged between: (i) Tot-Mg ≤ 0.70 and 0.81 ≤ Tot-Mg ≤ 0.90 for cf-PWV (*p* = 0.039); (ii) Tot-Mg ≤ 0.70 and Tot-Mg ≥ 0.91 for o-PWV (*p* = 0.046). We also found a statistically significant difference among groups of Ion-Mg values for the 24 h extremes of systolic blood pressure (*p* = 0.048) and among groups of Tot-Mg for BMI (*p* = 0.050). Females showed significantly lower levels of total magnesium (*p* = 0.035) and ionized magnesium (*p* < 0.001) than males. The overall agreement between magnesium analysis methods was 64% (95%CI: 60.8–67.7%). Our results confirm that Ion-Mg compared with Tot-Mg offers a different profile in detecting both correlations with hemodynamic and body composition parameters and dysmagnesemias. Lower levels of magnesium were associated with worse arterial aging parameters, larger 24 h blood pressure excursions, and higher BMI. Ion-Mg was superior in detecting the correlation with blood pressure only. Considering Ion-Mg as a more specific marker of the magnesium status, and the partially contradictory results of our explorative cross-sectional study, to avoid confounding factors and misinterpretations, ionized magnesium should be used as reference in future studies.

## 1. Introduction

Being the fourth most abundant electrolyte and the second most abundant cation in the intracellular compartment, magnesium (Mg) plays an important role in regulating many cellular biochemical processes and physiological functions in the human body [1,2,3]. In the brain, for instance, it influences receptor excitability, synaptic transmission, and neuronal plasticity [2,4]; at the neuromuscular junction, it regulates neurotransmitter release, action potential conduction, and muscle contraction [5]; in the cardiovascular system, instead, it helps in controlling myocardial contractility and the vascular tone [1,2,3].

Approximately 53% of total body magnesium is stored in the bone, 27% in the muscle, and approximately 19% in soft tissues, whereas only 1% is in the extracellular compartment. Serum magnesium may also be found either protein bound (~20%), complexed with anions (~15%), or ionized (~65%), the last of which constitutes the electrophysiological active form [6,7,8]. All of the abovementioned factors make the estimation of magnesium status quite challenging and, for practical reasons, even if ionized magnesium, total red blood cell magnesium, and the result of a magnesium loading test could offer a more accurate estimation of the magnesium status, everyday medicine usually relies on the determination of total serum magnesium (Tot-Mg) [9]. In addition, there is no overarching consensus regarding the exact range of adequate serum magnesium levels, so that different normality intervals have been suggested, both for research and clinical purposes. As for Tot-Mg, 0.75–0.95 mmol/L [10], 0.70–1.00 mmol/L [11], and 0.70–1.10 mmol/L [12] ranges have been frequently used, while for ionized magnesium (Ion-Mg), mainly 0.53–0.67 mmol/L [13] and 0.44–0.59 mmol/L [14] intervals have been applied.

Appropriate Mg levels are maintained by a fine balance between dietary intake, intestinal absorption and renal excretion, and the Mg shift from the extracellular to the intracellular spaces [1,3]. When this equilibrium fails, hypomagnesemia or, less commonly, hypermagnesemia ensue. Moreover, in the general population, hypomagnesemia has often been linked to risk factors such as excess body mass [15,16], older age [3,17], and use of medications, particularly proton pump inhibitors and diuretics [18,19].

Borderline alterations of magnesium levels may remain asymptomatic in a significant number of cases [20]. Rarely, severe magnesium toxicity could lead to serious complications, some of them being loss of deep tendon reflexes; flaccid paralysis; respiratory depression; hypotension; prolongation of PR, QRS, and QT intervals; bradycardia; even cardiac arrest [1,2,21,22,23,24,25]. Life-threatening conditions resulting from severe hypomagnesemia, on the other hand, may include tetany, seizures, coma, ventricular arrhythmias up to torsades de pointes, ischemic heart disease, and congestive heart failure [2,4,9,26,27]. 

Despite the recognized physiological implications of Mg in the homeostasis of the cardiovascular system, there is still conflicting evidence concerning its impact on the insurgence of coronary artery disease, cardiovascular diseases [28], and hypertension [29,30,31,32,33]. Concerning hypertension, presumably via the effect of magnesium on vascular tone and of the increase in angiotensin II, plasma aldosterone, and vasoconstrictive prostaglandins observed in hypomagnesemia, magnesium intake and blood pressure inversely correlate [9]. Furthermore, extracellular magnesium inhibits calcium influx into the cells via calcium channels and has vasodilatory effects on arterial smooth muscle cells [9]. Concerning cardiovascular diseases, low levels of magnesium have been associated with dyslipidemia, increased cellular oxidative stress, endothelial dysfunction, and platelet activation, all known risk factors [9].

Little is known about the relationship between serum Mg and arterial stiffness, although the mechanisms at the origin of the correlation with cardiovascular risk and hypertension previously cited are likely to also influence it [34,35,36,37,38]. Confirming the relationship, a past study showed a reduction in arterial stiffness (using pulse wave velocity as a parameter) in overweight and obese adults when supplementing daily magnesium for 24 weeks [35]; the literature regarding this topic, however, remains scant.

Our study aimed at exploring the correlations of magnesemia with PWV, blood pressure parameters, and body composition in the general population, investigating possible advantages of using ionized Mg instead of total magnesium in terms of the ability to highlight associations with pathological states.

## 2. Materials and Methods

The present study was based on a cross-sectional analysis of a sample of general population residents in southern Switzerland (Ticino), carried out in the years 2017 and 2018. Recruitment was based on a random sampling method via a mailing list provided by the Swiss Federal Statistical Department. Briefly, the study aimed to explore vascular aging and cardiovascular risk factors in adults [39,40].

From the initial population of 1202 individuals, only those in whom both total and ionized magnesium had been measured were included in the present analysis. Unfortunately, due to the temporary unavailability of the dedicated ionometer during the recruitment phase of the original study, ionized Mg testing was not attainable for 369 participants. In addition, PWV values were missing for 78 participants; therefore, 755 individuals were enrolled in the final sample (Figure 1). Potential differences between included and excluded patients were analyzed (Table A3).

For the measurement of ionized magnesium, as said before, an ionometer (*Microlyte 6 Analyzer*, KONE Instruments, Espoo, Finland) was used, while for total magnesium we applied the photometric method (Xylidyl blue in alkaline solution) via a *Cobas 8000* (Roche Diagnostics, Rotkreuz, Switzerland) device. Pulse wave velocity (PWV) measurements were obtained via two different methods and devices: oscillometric (*Mobil-O-Graph*, Industrielle Entwicklung Medizintechnik und Vertriebsgesellschaft, Stolberg, Germany; brachial pulse wave analysis; o-PWV) and tonometric (*SphygmoCor,* ACTOR, CardieX Limited, Sydney, Australia; carotid-femoral pulse wave determination; cf-PWV) [40]. Carotid-femoral PWV measurements were conducted on the patient’s dominant side in the supine position, after 10 min of rest. Participants were instructed to abstain from caffeine and tobacco use for four hours before the examination. The pulse wave path length was estimated by the software by entering the distance between both the carotid and the femoral artery and the supra-sternal notch [40].

The 24 h ambulatory blood pressure monitoring (ABPM) was obtained by the same device performing the oscillometric PWV measurement. Impedance parameters were obtained with a BIA impedance analyzer (*BIA 101*, Akern Bioresearch, Firenze, Italy), and the body composition was estimated using the software *Bodygram Plus* proposed by the same company [40].

Descriptive statistics were reported as the mean (±SD) and median (range) for quantitative variables and as frequencies and percentages for qualitative variables.

After descriptive statistical analysis of total magnesium and ionized magnesium, we divided both variables into four categories (i.e., hypomagnesemia, low-normal values, high-normal values, and hypermagnesemia) using the approach for frequency distribution construction as suggested by Witte and Witte [41]. Alternative total magnesium and ionized magnesium lower limits, respectively, set at 0.75 (Table A4) and 0.50 mmol/L (Table A6), were also investigated.

Comparisons among groups were performed through one-way ANOVA followed by Bonferroni post hoc analyses. ANCOVA was performed to account for PWV values adjusted for age. Correlations were analyzed through Pearson’s correlation coefficient. Normal distribution was verified through the skewness and kurtosis test and graph analysis. Linear spline regression was used to investigate a possible nonlinear relationship between total magnesium and Cf-PWV and o-PWV using the predetermined knots according to the previous classification. Statistical significance was set at 5% (*p* ≤ 0.05). Statistical analyses were performed through STATA17 (StataCorp., College Station, TX, USA).

## 3. Results

### 3.1. Patient’s Characteristics

Seven hundred and fifty-five patients were included in the study (435 females and 320 males) with a mean age of 54 (SD: 13; median: 54; range: 21–91) years. Patients in the healthy weight range (18.5 ≤ BMI ≤ 24.9) were 53.4% of the total, 31.9% (n = 241) were in a pre-obesity state (25.0 ≤ BMI ≤ 29.9), and 11.9% (n = 90) were obese (BMI ≥ 30.0). No statistical differences between included and excluded patients, in terms of sex, age, BMI, and hemodynamic parameters, were demonstrated (Table A3).

Total magnesium spanned from 0.62 to 1.09 mmol/L, with a mean of 0.83 (SD ± 0.06) mmol/L. Ionized magnesium ranged between 0.41 and 0.66 mmol/L with a mean of 0.53 (SD ± 0.30) mmol/L. Approximately 58.1% of the patients had high-normal magnesium values (0.81 ≤ Mg ≤ 0.90 mmol/L), followed by 29.7% of patients with low-normal magnesium values (0.71 ≤ Mg ≤ 0.80 mmol/L), 1.3% with hypomagnesemia (Mg ≤ 0.70 mmol/L), and 10.9% with hypermagnesemia (Mg ≥ 0.91 mmol/L), as determined through total magnesium. Similar percentages emerged also using ionized magnesium, according to which 52.8% (n = 399) of the patients had high-normal ionized magnesium values, 26.4% (n = 199) had low-normal ionized magnesium values, and, respectively, 1.3% (n = 10) and 19.5% (n = 147) had hypo- or hypermagnesemia. The mean difference between total and ionized magnesium was 0.31 (SD ± 0.04) mmol/L.

Females showed significantly lower levels of total magnesium and ionized magnesium than males (total magnesium 0.83 ± 0.06 vs. 0.84 ± 0.06 mmol/L, *p* = 0.035; ionized magnesium 0.52 ± 0.03 vs. 0.53 ± 0.04 mmol/L, *p* < 0.001).

Comprehensive patients’ characteristics are reported in Table A1.

### 3.2. Magnesium and PWV

Statistically significant differences emerged between groups defined according to ionized magnesium values for PWV obtained by SphygmoCor (cf-PWV) (Figure 2b and Table A2) and PWV obtained by Mobil-O-Graph (o-PWV) (Figure 2d and Table A2). The highest values of cf-PWV were observed in patients with Ion-Mg ≤ 0.45 mmol/L and in those with Ion-Mg ≥ 0.56 mmol/L, with statistically significant differences (*p* < 0.05) between patients with high-normal (7.4 ± 1.6 m/s) and low-normal values (7.0 ± 1.6 m/s) and between those with low-normal values and those with hypermagnesemia (7.6 ± 1.6 m/s) (Figure 2b). After adjusting cf-PWV for age, statistically significant differences were, however, seen only between patients with Tot-Mg ≤ 0.70 and the upper levels (Figure 2a). The same trends were observed for o-PWV (Figure 2c,d), although weak, statistically significant associations were noted between cf-PWV and o-PWV and total and ionized magnesium, respectively (Figure A2).

Analyzing the relationship between total magnesium and cf-PWV and o-PWV, a nonlinear relationship was confirmed by spline regression (Figure A1), showing a statistically significant decrease in cf-PWV (*β* = −33.8, SE = 13.9, *p* = 0.015) for total magnesium lower than 0.75 mmol/L, while a borderline, not statistically significant decrease appeared for o-PWV with total magnesium values lower than 0.75 mmol/L (*β* = −27.4, SE = 14.6, *p* = 0.060).

### 3.3. Magnesium: 24 h ABPM Minimum and Maximum Values

A statistically significant difference among groups defined according to the magnesium values for the delta between maximum and minimum systolic blood pressure in the 24 h ABPM, emerged using the ionized magnesium only (*p* = 0.048) (Figure 3b). Although the two-by-two group comparison was not significant, a trend towards higher values in patients with hypomagnesemia and hypermagnesemia was seen (Table A2).

### 3.4. Magnesium, BMI, and Body Composition

The correlation between BMI and the groups defined according to the magnesium values showed a significant statistical difference but only using the total magnesium as a parameter (Figure 4a). No statistically significant differences were instead observed for fatty mass (FM) and fatty free mass (FFM) parameters (Table A2).

### 3.5. Patients’ Reclassification Using Ionized Magnesium

Comparing the two magnesium analysis methods, ionized on whole blood and total on serum, we observed an overall agreement of 64.2% (95% CI: 60.8–67.7%); see Table 1 for details. A significant correlation was observed between total magnesium and ionized magnesium values (***ρ*** = 0.7428, 95% CI: 0.709–0.773 *p* = 0.0001) (Table A3).

### 3.6. Correlations between Blood Mg Levels and Measured Parameters under Alternative Intervals

Correlations between measured parameters (i.e., age, sex, BMI, FFM, FM, blood pressure, and PWV) and magnesium levels confirmed the trends previous described but without significant differences, using 0.75–0.85 mmol/L as the reference range for total magnesium (Table A4 and Table A5) and 0.50–0.54 mmol/L as the reference range for ionized magnesium (Table A6 and Table A7).

## 4. Discussion

Magnesium has been recognized as a relevant electrolyte for the physiology and homeostasis of the human body [12]; however, still today, its implications in the daily medical practice has not received its due recognition compared to other electrolytes such as sodium, potassium, and calcium [4]. Additionally, the biologically active form of Mg (ionized or free Mg) could be a better estimate of magnesium imbalances given its independence from any variations in serum-binding protein and anion complexing levels [2,42,43]. However, it is not routinely measured either in clinical or research settings, mainly due to the higher costs of ion-selective electrodes. The often silent presentation of both hypomagnesemic and hypermagnesemic conditions and the lack of standardized laboratory Mg ranges further challenge the efficient evaluation of patients’ magnesium status [11].

The literature has so far suggested the presence of a correlation between serum magnesium levels in the upper part of the distribution and lower arterial stiffness parameters [44], an inverse relationship between magnesium intake and blood pressure [9], and a higher risk of magnesium deficiency in overweight conditions [15]. Nonetheless, evidence regarding the role of Mg in cardiovascular diseases, including hypertension [29,30,31,32,33] and arterial aging parameters [34,35,36,37], and its correlation with body composition, on the one hand remains inconclusive, and on the other, derives mostly from total serum magnesium analyses.

With the aim of comparing the correlation profile of total and ionized magnesium with hemodynamic and body composition parameters, we exploratively examined the data of 755 subjects randomly chosen from the Southern Swiss general population, who participated, in the context of a previously described cross-sectional study [39], in a health assessment including serum total and whole blood ionized magnesium analyses, PWV, and 24 h BP and bioimpedance measurements. In order to investigate correlations, the sample group was divided into four equivalent classes according to magnesium levels, with most of the subjects within normal ranges. Interestingly, females showed significantly lower magnesium levels than males (Tot-Mg, *p* = 0.035; Ion-Mg, *p* < 0.001).

As far as carotid-femoral PWV is concerned, the results of the initial analysis showed statistically significant differences among magnesium classes, in particular between patients with high-normal and low-normal values and between those with low-normal values and magnesium levels above the upper limit. Higher values of PWV were seen both in patients with hypomagnesemia and in those with hypermagnesemia, suggesting a U-shaped nonlinear relationship (Figure A1). However, correcting PWV values for age, the numerical differences appeared to be less substantial, and the significance profile changed in favor of the differences between the total magnesium instead of ionized magnesium categories (Figure 1). The effect of the correction for age was explained by the fact that magnesemia has the tendency to decrease with age, and PWV, on the contrary, to increase. The oscillometric brachial PWV (o-PWV), often employed as a user-friendly alternative to the gold standard, cf-PWV, although known to be less specific and sensitive, showed a similar trend. As suggested in the literature, the negative effect of low magnesium levels on arterial stiffness could be explained by the consequences on arterial smooth muscle cells and on atherosclerotic processes of magnesium depletion.

Concerning blood pressure values, also influenced by the abovementioned pathophysiological mechanisms, an analogous behavior, statistically significant for ionized magnesium only, was seen analyzing the delta between the highest and lowest systolic 24 h ambulatory blood pressure monitoring values. These data could be of further interest considering the known association of blood pressure variability with the incidence of cardiovascular diseases and worse outcomes [45].

The BMI, on the contrary, correlated, among magnesium classes, with total magnesium only; while detailed body composition analysis using fatty and free fatty masses estimated by bioimpedance did not show any significant association (Table A1). In the three subanalyses, the different correlation profiles of total and ionized magnesium could be explained by confounding factors, such as protidemia, possibly related to nutrition and influencing the total magnesium results only.

The statistical analysis for all of the measured parameters was repeated using alternative normality intervals mentioned in the literature for both ionized and total magnesium, but we did not highlight a better correlation profile (Table A4, Table A5, Table A6 and Table A7).

The independent detection ability for low or low-normal magnesium levels of ionized compared with total magnesium was highlighted, reclassifying subjects according to the laboratory method used for the analysis (Table 1). In the context of an overall agreement between methods of 64%, 12% of the subjects classified in the “high-normal” range using Tot-Mg were reclassified into the “low-normal” range employing Ion-Mg, while 34% of subjects with “low-normal” values were reclassified into the “high-normal” range. Uneven frequencies of hypermagnesemic subjects also emerged when measuring total magnesium (n = 82) and ionized magnesium (n = 147), which could be justified by the choice of the normality intervals; the upper limit for ionized Mg being proportionately lower compared to total Mg. These findings suggest that the upper limit of ionized Mg used in most laboratories probably needs to be adjusted.

We can hypothesize that the higher sensitivity of ionized magnesium for magnesium imbalances allowed for the detection of the correlation with the amplitude of systolic blood pressure excursions, not confirmed using total magnesium and, on the opposite, for the unmasking of correlations generated by confounding factors.

## 5. Limitations

Although this cross-sectional study was carried out using a large sample size, we have to mention some limitations. First, being carried out in an unselected general population, we included few patients with hypomagnesemia as defined for both total and ionized magnesium. This was to be expected, since the prevalence of hypomagnesemia is estimated to be less than 2% in the general population [2,4], and our intent was explorative. Second, the impact of potential confounders outside the variables examined in the analysis was not considered. Third, selection bias could have occurred given that only the subjects of the original study population with ionized magnesium results were selected for this subanalysis. Fourth, the highlighted correlations among parameters could have been influenced by the chosen total and ionized Mg ranges, although they are supported by the literature. Finally, the statistical ability to detect correlations could have been influenced by possible nonlinear relationships between magnesium levels and hemodynamic and bioimpedance parameters.

## 6. Conclusions

In conclusion, our results confirm that ionized magnesium compared with total magnesium offers a different profile in detecting both correlations with hemodynamic and body composition parameters and dysmagnesemias. Lower levels of magnesium were associated with worse arterial aging parameters, larger 24 h blood pressure excursions, and higher BMI. Ionized magnesium was superior in detecting the correlation with blood pressure only. Considering ionized magnesium as a more specific marker of the magnesium status and the partially contradictory results of our explorative cross-sectional study, to avoid confounding factors and misinterpretations, ionized magnesium should be used as a reference in future studies on these topics.

## Figures and Tables

**Figure 1 jcm-11-04015-f001:**
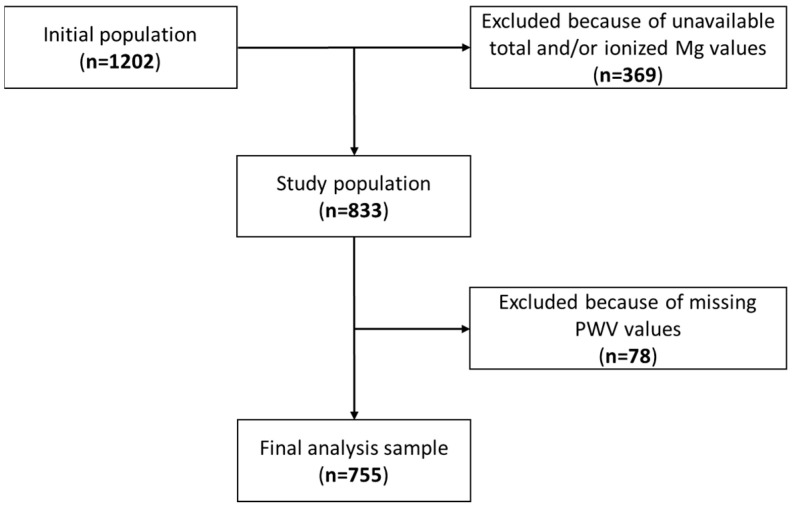
Flowchart showing the selection procedure of the participants.

**Figure 2 jcm-11-04015-f002:**
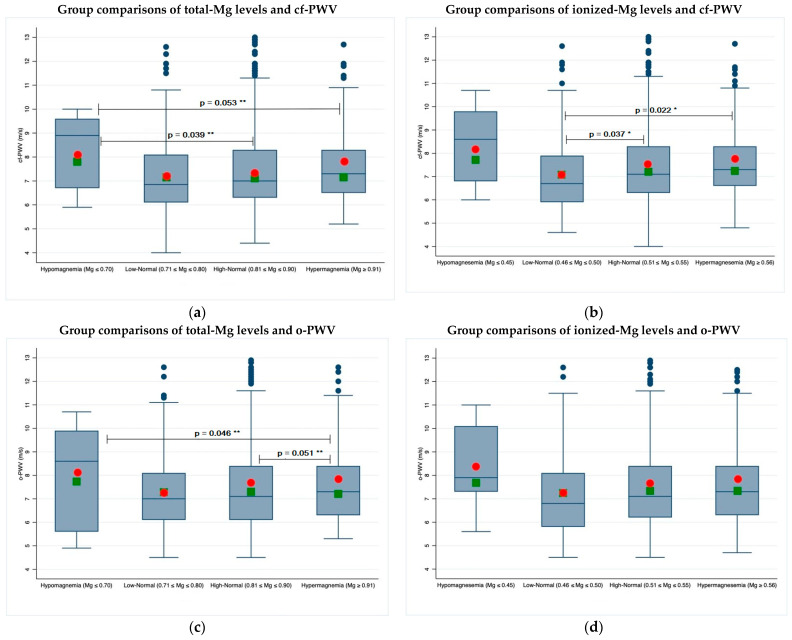
Two-by-two group comparisons of total and ionized magnesium levels and PWV: (**a**) cf-PWV and tot-Mg; (**b**) cf-PWV and ion-Mg; (**c**) o-PWV and tot-Mg; (**d**) o-PWV and ion-Mg. In the box plots, red circles and green squares indicate unadjusted and age-adjusted mean values, respectively, while the blue lines indicate the median values. Significant *p*-values between unadjusted (*) and adjusted (**) mean values are superimposed on the graphs.

**Figure 3 jcm-11-04015-f003:**
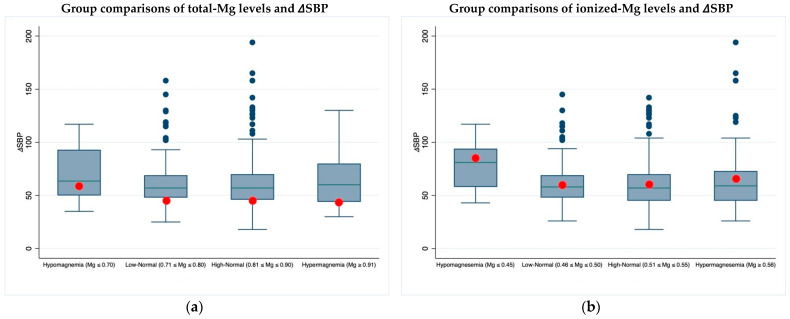
Two-by-two group comparisons of total and ionized magnesium levels and 𝛥SBP: (**a**) 𝛥SBP and tot-Mg; (**b**) 𝛥SBP and ion-Mg. In the box plots, red circles indicate mean values, while blue lines indicate median values.

**Figure 4 jcm-11-04015-f004:**
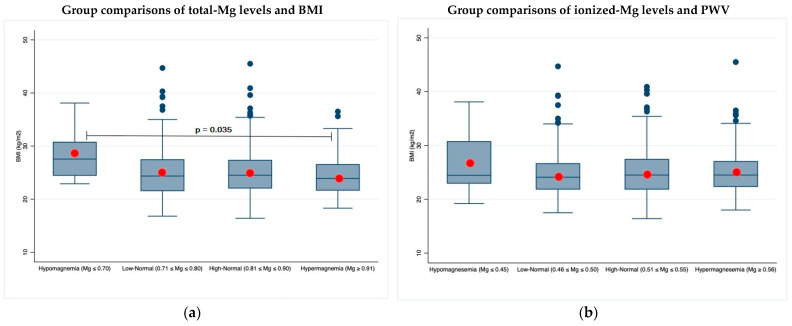
Two-by-two group comparisons of total and ionized magnesium levels and BMI: (**a**) BMI and tot-Mg; (**b**) BMI and ion-Mg. In the box plots, red circles indicate mean values, while blue lines indicate median values. Significant *p*-values between unadjusted mean values are superimposed on the graphs.

**Table 1 jcm-11-04015-t001:** Comparisons between total magnesium and ionized magnesium.

	Ion-Mg ≤ 0.45	0.46 ≤ Ion-Mg ≤ 0.50	0.51 ≤ Ion-Mg ≤ 0.55	Ion-Mg ≥ 0.56	Total
**Tot-Mg ≤ 0.70**	5	5	0	0	10
**0.71 ≤ Tot-Mg ≤ 0.80**	5	139	76	4	224
**0.81 ≤ Tot-Mg ≤ 0.90**	0	53	292	94	439
**Tot-Mg ≥ 0.91**	0	2	31	49	82
**Total**	10	199	399	147	755

Agreement: 64.2% (95% CI: 60.8–67.7%). Expected Agreement: 39.7% (95% CI: 34.1–45.4%). Ion-Mg, ionized magnesium; Tot-Mg, total magnesium.

## Data Availability

Data can be obtained from the authors upon reasonable request.

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
