# Peer review of "Does Ionized Magnesium Offer a Different Perspective Exploring the Association between Magnesemia and Targeted Cardiovascular Risk Factors?"

_jcm, 2022, doi:10.3390/jcm11144015_

Round 1
Reviewer 1 Report
This is an analysis of a large general population study from Switzerland. The analysis provides interesting results and manuscript is well-written and methodologically sound. However, I have a few comments that I think may help to improve the manuscript.
Page 1: In the abstract, it would be helpful to underpin the associations with the corresponding numbers.
Page 3: Figure 1: Please specify “other major data” in more detail.
Page 3, lines 127-130: In addition to analyzing the categorized variables for total and ionized Mg, it would be important to also analyze the continuous Mg variables.
Page 3, lines 133-134: Although the sample size was large, it is still necessary to inspect the distribution of the variables of interest for normality.
Page 4, lines 144-145: Specifically, please double-check the distribution of ionized Mg.
Pages 4-5, Figure 2: It would be helpful to state the specific measurement of PWV directly in the Figures, i.e., cf-PWV and o-PWV, as it makes it easier for the reader. In addition, it would be helpful to state in the Figures whether total or ionized Mg was analyzed.
Pages 4-5, Figure 2: Please, report the exact p-values in the Figures and in the text rather than “<0.05”.
Page 5: The authors provide Figures for all results described, but not for the result for 24-hour ABPM in paragraph 3.3. It would be helpful to also provide a Figure for these analyses.
Page 6, lines 189-192: Please provide 95% confidence intervals for Cohen’s Kappa and the correlation coefficient.
Page 6: The authors mention the U-shaped association between Mg and cf-PWV. As mentioned above, studying Mg as continuous instead of categorical variable could provide valuable additional information and could also help to better understand the shape of association (e.g. using splines).
Table A3: Please report the P-values for these correlations in more detail. Also, please provide 95% confidence intervals for Pearson correlation coefficients. Plotting the correlation coefficients using a heat map would also be helpful for the reader.
Page 7, lines 268-270: The authors mention the large number of individuals with missing data on Mg. Did the authors investigate whether there were any differences in the group of individuals with missing data on Mg and the group of individuals with data on Mg. Furthermore, did the authors consider imputing missing data?
The authors may consider investigating whether there are sex-differences in the relationship between total and ionized Mg and PWV.
Reviewer 2 Report
The manuscript presents a difference between total Mg and ionized Mg in correlation and clinical parameters. Despite interesting results, some improvement should be done:
1. Details about the study design and the main characteristics of the population should be presented.
2. I suggest to perform analysis using different reference values of Mg and iMg. Those applied seem to be quite low. Several and recent papers suggest to use at least 0.75 mg/dl or even 0.85 mg/dl, as lower reference value. Such an additional analysis will add more value to this study.
3. What is the reason of differences between frequency of hypermagnesemic subjects when measuring tMg and iMg?
Round 2
Reviewer 1 Report
The authors have addressed the comments adequatly, which has improved the manuscript significantly.
Reviewer 2 Report
After Revision article is suitable